# What Is the Evidence Globally for Culturally Safe Strategies to Improve Breast Cancer Outcomes for Indigenous Women in High Income Countries? A Systematic Review

**DOI:** 10.3390/ijerph18116073

**Published:** 2021-06-04

**Authors:** Vita Christie, Debbie Green, Janaki Amin, Christopher Pyke, Karen Littlejohn, John Skinner, Deb McCowen, Kylie Gwynne

**Affiliations:** 1Faculty of Medicine Health and Human Sciences, Macquarie University, Sydney 2109, Australia; janaki.amin@mq.edu.au (J.A.); kylie.gwynne@mq.edu.au (K.G.); 2Poche Centre for Indigenous Health, The University of Sydney, Sydney 2006, Australia; john.skinner@sydney.edu.au; 3Armajun Aboriginal Health Service, Armidale 2350, Australia; dgreen@armajun.org.au (D.G.); dmccowen@armajun.org.au (D.M.); 4Foundation for Breast Cancer Care, South Brisbane 4101, Australia; c_pyke@mc.mater.org.au (C.P.); karen@littlejohns.com.au (K.L.)

**Keywords:** breast cancer, aboriginal, indigenous, cultural safety, indigenous health

## Abstract

The aim was to systematically assess the evidence on whether cultural safety affects breast cancer outcomes with regards to care for Indigenous women in high income countries. We conducted a systematic review in accordance with PRISMA guidelines of peer-reviewed articles in Medline, EMBASE, CINAHL, Scopus, Web of Science, Proquest Sociology and Informit Rural health database and Indigenous collection databases. Key inclusion criteria were: adult female patients with breast cancer; high income country setting; outcome measure, including screening, diagnosis, treatment and follow up care. A total of 15 were selected. We developed a Community Engagement assessment tool in consultation with aboriginal researchers, based on the National Health and Medical Research Councils’ community engagement guidelines, against which studies were appraised. This novel element allowed us to evaluate the literature from a new and highly relevant perspective. Thematic analysis of all 15 studies was also undertaken. Despite limited literature there are evidence-based strategies that are likely to improve outcomes for Indigenous women with breast cancer in high income countries and indicate that culture makes a positive difference. It is also clear that strong Indigenous community leadership and governance at all stages of the research including design is an imperative for feasibility.

## 1. Introduction

Breast cancer is the most commonly diagnosed cancer in Australia, representing 28% of all cancer incidence in women and the second highest number of deaths [1]. Mortality due to breast cancer has declined significantly over recent decades. This coincides with improved rates of early detection following introduction of national population-based mammography screening programs [2], yet Aboriginal and Torres Strait Islander women in Australia continue to face high mortality rates, despite an incidence of breast cancer on par with or less than non-Indigenous women [1]. Between 1998 and 2013, there has been no significant decrease in the Aboriginal mortality rates for breast cancer [1]. When looking at the overall picture, tumour pathology is shown to play only a minimal role in the disparity of survival outcomes as compared with preventable causes relating to delayed diagnosis and treatment [3].

Uptake of screening has increased over time among both non-Indigenous and Indigenous populations in Australia, however a significant gap remains. Australia’s national population-based screening program BreastScreen offers free 2-yearly mammograms targeting women aged 50–74 years. In 2019, 41% of Aboriginal and Torres Strait Islander women in this age group participated compared with 54% of non-Indigenous women [4]. As a consequence of the gap, Aboriginal and Torres Strait Islander women are more likely to be diagnosed at an advanced stage, experiencing worse disease outcomes and lower rates of survival [5].

While age is the greatest risk factor for breast cancer, Aboriginal women are more likely to be younger than non-Aboriginal women at the time of diagnosis [6]. Aboriginal women are more likely to receive more invasive surgical treatment compared with their non-Aboriginal counterparts at the same breast cancer stage [7]. This is likely due to perception of difficulty in engaging Aboriginal women for regular and timely follow-up monitoring and care.

There is evidently a demonstrable need for improved screening, diagnostic and care pathways for Aboriginal women in Australia [8]. The literature identifies numerous barriers and enabling factors which contribute to ease of access, timeliness, and quality of care for Aboriginal women with regard to breast cancer screening and services [9,10,11,12,13,14,15,16,17].

Overwhelmingly, contributing factors are related to the lack of cultural safety within health services. Cultural safety has been defined as “*An environment that is safe for people: where there is no assault, challenge or denial of their identity, of who they are and what they need. It is about shared respect, shared meaning, shared knowledge and experience, of learning, living and working together with dignity and truly listening*” [18]. Aboriginal women are apprehensive about utilising services due to recent or historical experiences of racism, lack of culturally safe care and a deficit of resources featuring culturally appropriate educational and health promotion messages. Initiatives which focused on resourcing community-led initiatives to raise awareness facilitated increased uptake and provided culturally safe care. This care involved aboriginal health workers and highlights the importance of primary health care following diagnosis [19]. Furthermore, individuals were less likely to engage in services as a consequence of previous experiences or the experiences of women they knew with mammography and breast cancer. Feelings of shame or stigma were also cited as was the impact of financial barriers and geographical remoteness.

There is a body of evidence surrounding initiatives aimed at increasing breast screening among Aboriginal women which indicates that success is highest where there are partnerships with Aboriginal community-controlled organisations [16]. These initiatives implemented culturally appropriate engagement strategies, such as involving community members in planning and implementing evidence to address a range of social, cultural, personal and economic factors. An expanding evidence base supports the use of ‘co-design’ as a research methodology for the design, implementation, and evaluation of successful, cost-effective and sustainable strengths-based solutions to health challenges among Aboriginal communities [20].

This review aims primarily to examine the role of culture when it comes to improving Indigenous women’s health outcomes, and secondarily to elucidate factors predicting success, and barriers to success of programs.

## 2. Methods

### 2.1. Study Design

This study is a systematic review of peer-reviewed literature and is reported using the Preferred Reporting Items for Systematic Reviews and Meta-Analysis (PRISMA) statement guidelines [21]. The study was registered with PROSPERO International prospective register of systematic reviews (CRD42019134757) [22].

### 2.2. Eligibility Criteria

Population: Indigenous women in the US, Canada, Australia, New Zealand, Norway and Japan participating in breast screening, diagnosis, treatment and follow up Intervention: Prevention, Screening, Diagnosis, Follow Up. Comparator: Non-Indigenous women in high income countries, as defined by the WHO. Outcome: Mortality, participation rates, adherence to guidelines, cultural safety (was any provided and did it make a difference?)

### 2.3. Search Strategy

Seven online databases were searched: MEDLINE, EMBASE, Scopus, Web of Science, Proquest Sociology, Cinahl and Aboriginal and Torres Strait Islander Health Informit database.

The strategy had three main concepts relating to: (1) Indigenous populations around the world in Australia, New Zealand, USA, Canada, Japan and Norway and related terms, (2) Breast cancer and related terms and (3) Culturally appropriate and related terms.

The search terms were: ((aboriginal OR indigenous OR “first nations” OR “torres strait” OR “first peoples” OR “Native american*” OR maori OR sami OR inuit OR ainu) AND ((cultural* W/3 (safe* OR appropriate* OR competen* OR suitab* OR inclusiv*)) OR culture) AND ((cancer OR tumo?r* OR neoplasm*)))

The terms were combined with limitations on human studies only and language was limited to English. Eligible studies were limited to journal articles, observational studies, clinical studies, and clinical trials. The search terms were adapted for use with other bibliographic databases in combination with database-specific filters, where these are available.

### 2.4. Study Selection Process

Following duplicate removal, the first author screened all studies retrieved using the search strategy and any additional sources using the title and/or abstract to determine inclusion based on the criteria outlined above. The full text of these articles was then independently assessed for eligibility by the first author and discrepancies were resolved by consensus between the first and Senior authors. Table 1 is a summary of the final 15 papers.

### 2.5. Community Engagement Assessment

This review applied the National Health and Medical Research Council (NHMRC) community engagement guidelines [23] and assigned scoring system to key components. Using these guidelines our studies were assessed by three members of the research team, heavily influenced by the Aboriginal researcher on the team for the meaning of the categories. The elements that are assessed broadly fall into the following categories: (1) issue identified by community; (2) Indigenous governance; (3) capacity building; (4) cultural consideration in design; and (5) respecting community experience. Each study was scored 1 for yes and 0 for no and the total was tallied to show where the study fell on the scale. Each paper was scored between 0 and 5.

### 2.6. Thematic Analysis

All final 15 studies selected for community engagement assessment underwent a thematic analysis. Each of the studies was analysed against these categories and evidence of themes was recorded. If a theme emerged in two or more papers, it was considered ‘strong’. The thematic analysis was conducted by one researcher (VC) and reviewed by a second (KG). The researchers reached consensus through discussion and subsequently sent to entire author team.

### 2.7. Data Extraction and Synthesis

General characteristics of the article, participants, interventions, study outcomes and measures were extracted by the first author using a purpose designed form. The first author then compared the results of the community engagement scores with the Thematic analysis.

## 3. Results

The initial search yielded 1714 records. After removing 701 duplicates 1013 records remained. During the title review 699 were excluded as they were either: biomedical, focused on complementary medicines and spirituality, not wealthy countries, animal studies, not in English language, males only, limited to other individual bodily system, regarding youth only. Thus 314 articles remained, which were then subjected to an abstract review, with further exclusions made: focused on family experience only, studies including non-Indigenous population groups, related to clinical trials only, discussion of HPV/Cervical cancer only, speaker abstracts, conference abstracts, PhD dissertations, prevention focus only, service provider focus, systematic reviews, reviews, no outcome data, all cancers, men and women, specific cancer(not breast) only, regarding challenges within the health system but not for community, format(such as book excerpt) or because they did not use primary data sources or were about cancer in general. 15 remained and were subject to the community engagement tool and a thematic analysis. This is shown in Figure 1 the PRISMA Flow Chart.

### 3.1. Community Engagement Tool

All 15 papers were assessed using the community engagement tool we developed based on the NHMRC Community engagement guidelines. The studies fell across the spectrum in community engagement scores. One study scored 5, eight scored 4, two scored 3, one scored 2 and the remaining three scored 0. The community engagement factors that were reported in the majority of studies were Cultural consideration (x/15) in design and Respecting community difference (y/15). The community engagement factor least likely to be reported was Issue identified by community (z/15). Table 2 shows the community engagement score for each of the papers and which specific elements of community engagement were reported, giving an indication of which elements take precedence over others. This indicates which of the five elements are used in each of the studies.

Besides the eight studies which scored between 4 and 5, two of the remaining papers scored 3, one scored 1 and 3 scored 0. This implies that, while the majority of projects have used community engagement as a focus of research, others may not have taken it into account at all. When looking at the studies that scored 0 it was notable that none explicitly mentioned Indigenous researchers working on the study and nor did they view this as a limitation of their studies [30,33,37]. It should be noted that not all studies may have *reported* details of community engagement despite this being a key component of successful study design with Indigenous communities.

### 3.2. Thematic Analysis

Four themes emerged strongly in the thematic analysis, these were: (1) Silence; (2) Service Access; (3) Cultural conception of cancer; and (4) Family and community support. The presence of the theme is shown in Table 3 These themes are explored in detail in the following Discussion section.

## 4. Discussion

This novel study demonstrates that if researchers, clinicians and community members work together, solutions exist for complex health issues like breast cancer prevention and treatment. Despite the small number of relevant studies in the review, the principles of co-design, including community engagement, community governance and capacity building are central to effective research with Indigenous communities [38,39]. In a 2005 review of US breast health outreach and education with native communities it was stated that “*the most successful and effective programs were those developed with input from the population and tailored to meet the needs of the region and specific tribes*” [40].

### 4.1. The Importance of Community Engagement

While only one of the studies scored five on the tool, a further eight scored four, indicating the importance of community engagement in doing publishable research with Indigenous peoples. The top scoring study attributed the success of the research to the focus on community engagement, stating that “*With an underlying framework of respect and communication, diverse skills and knowledge, a coalition was formed between community and external programs to guide the project and ensure that the community had ownership of both the problems and the solutions*” [29]. One of the studies scoring four stated that “*The involvement of community representatives, working alongside researchers, in baseline survey planning, helped assure the survey was acceptable to the participants and the community as a whole*” [24]. When observing similarities between the studies that scored 4–5 on the community engagement tool, there were two concepts that prevailed- (1) community participation and community being the researchers of their own health concerns and (2) incorporation of culture in research design, such as methods of recruitment and methods of data collection. As one article explains, “*The goal of Talking Circles is to make one’s self into a vessel to receive spirit through a series of processes, for example, purification of place, self, and community. The cultural and spiritual experience changes a person’s sensory and self-awareness* [25]. This speaks to the depth and complexity of cultural practices.

An integral element of community engagement is capacity building. Not only does it engage the community and help to build a culturally safe protocol, it also helps to ensure sustainability and increase health literacy. The top scoring study attributed a good deal of its success to the fact that capacity building was a focus of every part of the project, claiming that it “*was assumed to be a goal in itself, separate from programmatic objectives to improve screening rates*” [29].

### 4.2. Themes Emerging in the Literature

The thematic analysis revealed four themes across the literature included in this study. The first theme, Silence, remains a theme that needs further investigation as it appeared to sit across all the other themes, and not squarely in one. Silence appears both from the person with cancer and also those supporting them. In fact, in some Indigenous cultures there “*is no word for cancer*” [25]. While some of the literature attributes this to fear, it is also viewed in the light of denial, based on beliefs such as ‘cancer is a white person’s disease’, and misunderstanding. For example, cancer will not be openly discussed so as to avoid *inviting* it into the community [41]; or “*cultural views tend to discourage open verbal discussion about negative health topics because it is believed that talking and thinking about these topics in first-person rather than third-person, may bring disease or illness upon the person speaking*” [42]. Silence blurred the boundaries between acting as an obstacle to accessing services, being an ingrained belief about the disease and being a community attitude. That it increased the sense of isolation of the person with cancer as they felt they were not able to discuss it even with their closest family members and supporters.

Theme number two, Service Access, is much more easily delineated. There were several emerging sub-themes, such as specific barriers to access or delivery. One sub-there identified was a lack of appropriate education around the disease, one study stating that “*Many Navajo women avoid mammography screening, due in part, to a lack of knowledge and a sense of hopelessness associated with cancer*” [35]. Another study identified “increased education on breast cancer and screening, appropriate both to cultural differences and literacy levels, as the key facilitator to improve screening participation” [15]. Another sub-theme was a lack of culturally appropriate services or a mistrust of services. One study explained that the womens’ “*lack of trust in their providers and in Western Medicine made them not want to get screened at all*” [28]. The barriers to service access and delivery also included more practical considerations, such as financial, transport or lack of child care. One study identified that “*Preventive care is considered of tertiary priority and is available if there is money left over after other services are covered. In addition, patients do not often use available preventive services or come in when they are symptomatic from an illness*” [27] due to the barriers just mentioned. This theme helps us understand why Indigenous women present later than other women to cancer services and is an important consideration in service design. Engaging Indigenous women in service design to address issue of access, welcome and cultural safety are key considerations form the literature.

Theme number three, Cultural conception of cancer, covered a range of beliefs about and attitudes towards the disease and identified that beliefs about cancer can directly prevent women seeking screening, diagnosis or treatment [37]. While there is a lot of fear around breast cancer, there is also a belief that it will not happen to Indigenous peoples as it is a white person’s disease [31]. In one study “*Women articulated a conception of how cancer works, describing cancer as threatening and damaging, often final. Some saw cancer as worsening their economic and mental state of wellbeing*” [25] where as in a different study “*participants emphasized that many older Native women did not feel that they were even at risk for cancer, and they did not seek preventive testing*” [30]. Other somewhat contradictory beliefs identified were that cancer is viewed as “*punishment for one’s own actions or those of a family member, (or that) one needs to “wear the pain” of cancer to protect their community, cancer is a natural part of one’s path, cancer results from a curse or a personal violation of tribal mores, and there is a contagious “cancer spirit*” [31]. In one American Indigenous language, the word for cancer directly translates as ‘the sore that does not heal’ [43] speaking directly to a cultural understanding around the nature of cancer and how varied experiences can be. Furthermore, it stands in direct opposition to the idea that screening early can make a difference, despite this being a well-documented fact. This explains why the mortality rate for Indigenous peoples from cancer are far higher than non-indigenous despite having lower prevalence. It will be important for future studies to work with Indigenous women to develop, define and create language about cancer that is culturally acceptable, builds confidence in screening and treatment pathways and dispels some of the myths about intrinsic and extrinsic beliefs about the causes and impacts of cancer.

Theme number four, Family and community support, contained several sub-themes, including lack of proper dissemination of information, family and community support encouraging positive or change of behaviour and a general understanding of community of the difference between survival and non-survival. One study identified the motivator to screen as “*encouragement from significant others, particularly family, friends, elders, and health providers, who either advised keeping the scheduled women’s health exams and the mammography referral appointment or counselled keeping oneself healthy*” [25] and another observed that “*participants were comfortable in making an oral commitment to the family and community to engage in screening behaviors*” and that this knowledge added to the researchers’ understanding and “*provided more depth related to family involvement; women brought children, aunts, mothers, grandmothers, and spouses*” [36] to assist with learning about breast cancer. Beyond the family a buddy system or support from women in the community was suggested as a way to assist in understanding and feel support [15,28]. Alternatively, a negative reaction from spouse and partner was identified “*as a barrier to breast self-care*” [31].

Family and community not only offer the support but are also affected by the disease. One participant in a study said “*Cancer isn’t about me. It’s about us. Families and communities are affected, too*” [32]. and a different study identified “*A key driver for engagement in breast screening was the perception of investing in health for the sake of the next generation*” [15]. The experiences of family are fundamental to the person with cancer’s experience of the treatment process. Engaging family and community in the process both increases support for women with breast cancer and increases health literacy for all.

Once the themes had been established, the first author looked at whether there was a correlation between the community engagement score and the thematic analysis. Not being able to find a clear one, the first author looked at the studies that were high scoring in community engagement to distil the elements these studies had in common; inferring the factors that were reported as having an impact. This is shown in Table 4.

#### Culture Making a Difference

Overwhelmingly the literature finds that culture makes a real difference to Indigenous women when it comes to the breast cancer journey. Many studies found that women fared better when culture was incorporated into the screening, diagnosis, treatment and follow up care. Further, none of the literature claimed there was no relationship between culture and health. One study found that “*When poor experiences with mammography were discussed, not up to date women focused more on cultural insensitivity and lack of respect than the mammogram itself*” [36] indicating that it wasn’t overwhelmingly the physical discomfort/pain of the mammogram that was felt. Another study found that “*Increases in screening participation must be matched by culturally appropriate programs that deal with abnormal findings and facilitate Aboriginal women’s pathway through cancer treatment services. Only then will earlier detection of breast cancer translate into better outcomes*” [18].

Much of the literature found that the use of Indigenous methodologies [25] and Indigenous researchers was effective when communicating with Indigenous women. Not only could the Indigenous women relate to the people talking about breast cancer, this relationship “*sparked an interest in gaining more information about treatment options and reduced anxiety related to the treatment process*” [42]. A limitation of this research is the small number of studies included. Unfortunately, the vast majority of Indigenous health research is descriptive or epidemiological [44] and further research, including on breast cancer, is needed to ensure that health interventions are efficacious for Indigenous peoples.

## 5. Conclusions

This literature in this review found that culture makes a significant difference to Indigenous peoples in wealthy countries when it comes to cancer in general. When looking specifically at breast cancer for Indigenous women in high income countries, much the same message emerges on a smaller scale. The literature that is available about breast cancer overwhelmingly emerges from the US and Canada, compared with a relative paucity from the Australasian nations included in the search, however the concepts are the same.

From this review, it is clear that there needs to be more research about capacity building, culturally safe services and building health literacy. Based on this study we can infer that the impact of these elements is vast, but it is not clear just how integral they are to the success of services for Indigenous women with breast cancer.

The literature provides evidence-based strategies that are likely to improve outcomes for Indigenous women with breast cancer in wealthy countries. It is also clear that strong Indigenous community leadership and governance at all stages of the research including design is an imperative for feasibility.

## Figures and Tables

**Figure 1 ijerph-18-06073-f001:**
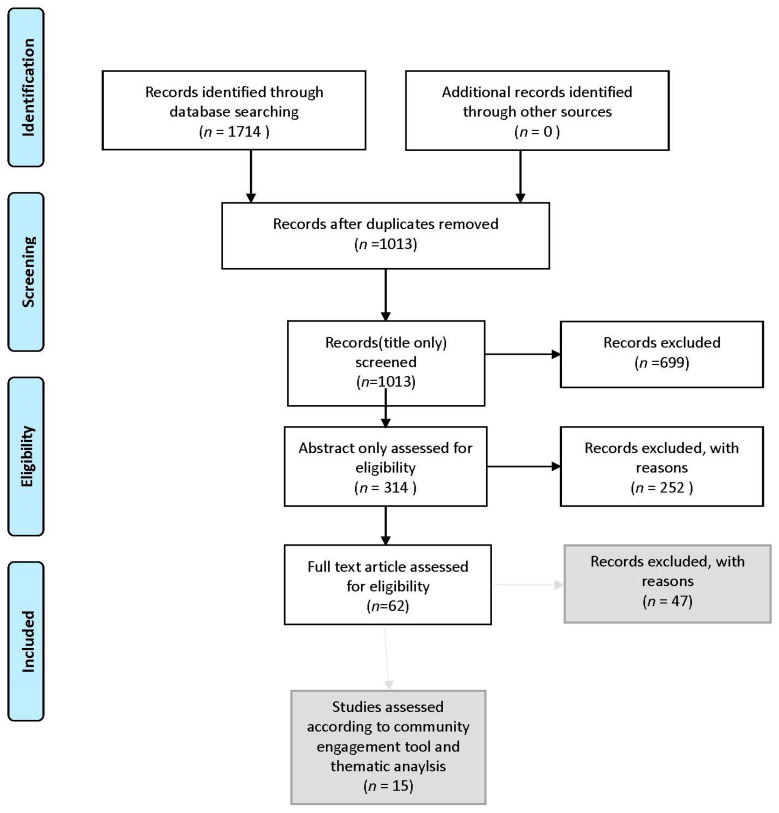
PRISMA flow diagram. The boxes with the grey background indicate the final selection process for the studies.

**Table 1 ijerph-18-06073-t001:** Summary of papers.

Title	Authors	Reference	Numbers	Methods	Location
A breast and cervical cancer project in a native Hawaiian community: Wai’anae cancer research project	Banner, R.O.	[24]	1260	Baseline telephone survey	Hawaii, USA
	DeCambra, H.				
	Enos, R.				
	Gotay, C.				
	Hammond, O.W.				
	Hedlung, N.				
	Issell, B.F.				
	Matsunaga, D.S.				
	Tsark, J.A.				
Talking circles: Northern Plains tribes American Indian women’s views of cancer as a health issue	Becker, S.A.	[25]	28	Talking Circle and focus group methodology, combined with Affonso’s Focus Groups Analytic Schema.	South Dakota, USA
	Affonso, D.D.				
	Beard, M.B.H.				
Impact of a community-based breast cancer screening program on Hopi women	Brown, S.R.	[26]	250	Community meetings, focus groups, and researchers jointly developed a culturally appropriate survey instrument.	Arizona, USA
	Nuno, T.				
	Joshweseoma, L.				
	Begay, R.C.				
	Goodluck, C.				
	Harris, R.B.				
	Brown, S.R.				
	Nuno, T.				
	Joshweseoma, L.				
	Begay, R.C.				
	Goodluck, C.				
	Harris, R.B.				
American Indian Community Leader and Provider Views of Needs and Barriers to Mammography	Daley, C.	[27]	30 totals (*n* = 13 community leaders; *n* = 17 health providers)	Interviews	Kansas, USA
	Filippi, M.				
	James, A.				
	Weir, M.				
	Braiuca, S.				
	Kaur, B.				
	Choi, W.				
	Greiner, K.				
Breast cancer screening beliefs and behaviors among American Indian women in Kansas and Missouri: a qualitative inquiry	Daley, C.M.	[28]	84	Focus groups	Kansas and Missouri, USA
	Kraemer-Diaz, A.				
	James, A.S.				
	Monteau, D.				
	Joseph, S.				
	Pacheco, J.				
	Bull, J.W.				
	Cully, A.				
	Choi, W.S.				
	Greiner, K.A.				
A socioecological approach to improving mammography rates in a tribal community	English, K.C.	[29]	25	Focus groups	New Mexico, USA
	Fairbanks, J.				
	Finster, C.E.				
	Rafelito, A.				
	Luna, J.				
	Kennedy, M.				
Assessing cultural sensitivity of breast cancer information for older Aboriginal women	Friedman, D.B.	[30]	25	Interviews	Ontario, Canada
	Hoffman-Goetz, L.				
Use of the Talking Circle for Comanche Women’s Breast Health Education	Haozous, E.A.	[31]	7	Talking Circle	Oklahoma, USA
	Eschiti, V.				
	Lauderdale, J.				
	Hill, C.				
Testing the feasibility of a culturally tailored breast cancer screening intervention with Native Hawaiian women in rural churches	Ka’opua, L.S.	[32]	198	randomized, two-group pre–post control group comparison	Hawaii, USA
	Park, S.H.				
	Ward, M.E.				
	Braun, K.L.				
Perspectives of Aboriginal women on participation in mammographic screening: a step towards improving services	Pilkington, L.	[15]	65	Semi-structured interviews, focus group discussions and yarning sessions	Western Australia
	Haigh, M.M.				
	Durey, A.				
	Katzenellenbogen, J.M.				
	Thompson, S.C.				
Breast cancer literacy and health beliefs related to breast cancer screening among American Indian women	Roh, S.	[33]	286	Self-administered survey	South Dakota, US
	Burnette, C.E.				
	Lee, Y.S.				
	Jun, J.S.				
	Lee, H.Y.				
	Lee, K.H.				
Breast cancer education for Navajo women: a pilot study evaluating a culturally relevant video	Sanderson, P.R.	[34]	40 in total (*n* = 14 women diagnosed with breast cancer; *n* = 26 healthcare providers)	Questionnaires	Arizona, US
	Teufel-Shone, N.I.				
	Baldwin, J.A.				
	Sandoval, N.				
	Robinson, F.				
Development and Evaluation of a Cancer Literacy Intervention to Promote Mammography Screening Among Navajo Women: A Pilot Study	Sinicrope, P.S.	[35]	25	Interviews	Navajo Nation, US
	Bauer, M.C.				
	Patten, C.A.				
	Austin-Garrison, M.				
	Garcia, L.				
	Hughes, C.A.				
	Bock, M.J.				
	Decker, P.A.				
	Yost, K.J.				
	Petersen, W.O.				
	Buki, L.P.				
	Garrison, E.R.				
Conducting a Feasibility Study in Women’s Health Screening Among Women in a Pacific Northwest American Indian Tribe	Strickland, C. June	[36]	10	Interviews	Pacific Northwest, US
	Hillaire, Elaine				
Predictors of regular mammography use among American Indian women in Oklahoma: a cross-sectional study	Tolma, Eleni L.	[37]	255	Survey	Oklahoma
	Stoner, Julie A.				
	Li, Ji				
	Kim, Yoonsang				
	Engelman, Kimberly K.				

**Table 2 ijerph-18-06073-t002:** Community engagement score and elements reported on.

Study	Issue Identified by the Community	Indigenous Governance	Capacity Building	Cultural Consideration in Design	Respecting Community Experience	Total Score
Banner et al. [24]		1	1	1	1	4
Becker et al. [25]		1	1	1	1	4
Brown et al. [26]	1	1		1	1	4
Daley et al. 1 [27]		1	1	1	1	4
Daley et al. 2 [28]		1	1	1	1	4
English et al. [29]	1	1	1	1	1	5
Friedman et al. [30]						0
Haozous et al. [31]			1	1	1	3
Ka’opua et al. [32]	1	1		1	1	4
Pilkington et al. [15]		1		1	1	3
Roh et al. [33]						0
Sanderson et al. [34]				1		1
Sinicrope et al. [35]		1	1	1	1	4
Strickland et al. [36]		1	1	1	1	4
Tolma et al. [37]						0
**Total number of studies**			**15**			

**Table 3 ijerph-18-06073-t003:** Summary of themes emerging from the literature.

Study	Theme 1: Silence	Theme 2: Service Access	Theme 3: Cultural Conception of Cancer	Theme 4: Family and Community Support
Banner et al. [24]				X
Becker et al. [25]		X	X	
Brown et al. [26]				X
Daley et al. 1 [27]		X	X	
Daley et al. 2 [28]		X	X	
English et al. [29]	X	X	X	X
Friedman et al. [30]		X		
Haozous et al. [31]		X	X	
Ka’opua et al. [32]		X		X
Pilkington et al. [15]		X		X
Roh et al. [33]		X	X	
Sanderson et al. [34]	X	X	X	
Sinicrope et al. [35]		X	X	
Strickland et al. [36]		X		X
Tolma et al. [37]		X	X	

**Table 4 ijerph-18-06073-t004:** The two most common engagement mechanisms.

Study	Success Factor 1: Community Participation; Researchers of Own Health Concerns	Success Fact 2: Incorporation of Culture in Research Design
Brown [26]	x	
Banner [24]		x
Becker [25]	x	x
Daley et al. 1 [27]	x	
Daley et al. 2 [28]	x	x
English et al. [29]	x	x
Ka’opua et al. [32]		x
Sinicrope et al. [35]		x
Strickland et al. [36]	x	x

## Data Availability

No datasets were generated or analysed during the current study.

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
