# Peer review of "What Is the Evidence Globally for Culturally Safe Strategies to Improve Breast Cancer Outcomes for Indigenous Women in High Income Countries? A Systematic Review"

_ijerph, 2021, doi:10.3390/ijerph18116073_

Round 1
Reviewer 1 Report
The authors conducted a systematic review of 15 studies from high-income countries around the globe with indigenous populations to investigate the impact of culture on cancer outcome in terms of screening, diagnosis, treatment and follow-up. Papers were reviewed and assessed on their community engagement using a novel scoring system and thereafter thematically analyzed. The papers with the highest scores in community engagement were further reviewed to discover the common elements.
Overall, this work has some notable strengths: the systematic review is fairly extensive encompassing 15 studies of Indigenous women populations from six high income level countries (US, Canada, Australia, New Zealand, Norway and Japan). I give more specific comments below:
Major concerns
The authors emphasize the importance of co-design, i.e., including community engagement, community governance and capacity building, in improving the breast cancer outcome in indigenous women globally. The author does this by reviewing how many of the chosen articles focus on community engagement and including culturally appropriate strategies. They further analyze the overarching themes of the papers. A natural question that arises is, how the papers review the strength of cultural safety. The author inferred the “factors for success” from the common elements in papers scoring high in community engagement. However, I believe it is imperative to the impact of this study that the success of the studies be quantified in a manner other than the scoring of community engagement. In the eligibility criteria, the outcome measures mentioned are “mortality, participation rates and adherence to guidelines”. These are quantifiable measures of success. Is there a correlation between the studies with e.g., low mortality rate, high participation rate and high adherence to guidelines, and the community engagement? Are the studies that are successful in terms of quantifiable outcome measures also the studies that show high community engagement?
It would be better to provide a concrete definition of cultural safety and culturally safe therapies, in the introduction. The idea can be found only after careful reading of the discussion.
Details
Title:
- The key focus point of the paper is not breast cancer therapy, i.e., treatment, thus I suggest rephrasing the title. Perhaps replace “therapies” with “strategies”.
- change “to improving” to “to improve”.
Abstract:
- The first sentence reads: “cancer outcomes with regards to screening, diagnosis, treatment and follow up”, however, the results section of this paper does not present any data on cancer screening, diagnosis, treatment or follow-up. I suggest rephrasing this.
- Line 24-25: “Thematic analysis of all 15 studies was also undertaken. A total of 15 were selected assessment of community engagement and the final 15 studies also underwent a thematic analysis” – these sentences have some redundancies and need to be rephrased.
- Line 28: “…clearly indicates that culture makes a positive difference” - this needs more justification, the statement seems pretty strong.
Introduction:
- Line 42: “tumour biology is shown to play only a minimal role in the disparity of survival outcomes as compared with preventable causes relating to delayed diagnosis and treatment”
- This paper cited looked specifically at histopathological characteristics of the tumors. Did it consider possible underlying genetic heterogeneity driving the tumor aggressiveness? If not, perhaps reword “tumor biology”.
- Line 77: Define “culturally appropriate engagement strategies”.
Methods:
- Line 92: “Indigenous women’s” should be “Indigenous women”
- Line 93: “participation” should be “participating”. Put period after “follow-up”.
- Line 95: consider changing “wealthy countries” into “high-income level countries” as defined by the WHO.
- Line 96: Again, how is cultural safety defined?
Results:
- Line 171: “This suggests that each of the five elements of the community engagement tool do not hold equal weighting.”: this is confusing and needs further clarity.
- Line 174: “Besides the studies which scored between 4 and 5, the remaining papers were concentrated around the scores of 0-1“. I would perhaps start off mentioning that the median and mode of the scores is 4. Seeing two papers have a score of 3, I would be careful with the statement that the scores concentrate around 0-1.
- Line 183: “Four themes emerged strongly in the thematic analysis, these were…”: Can the author clarify the benchmark for considering a strong theme? The theme “silence” emerges twice in the papers, thus was that the threshold? Perhaps elucidate in methods section.
Discussion:
- Silence as a theme: Have the authors considered grouping the “silence” theme and “cultural conception of cancer” theme together? It seems as though silence might be a product of the conception of cancer and that they are interwoven.
- Line 305: “not being able to find a justifiable one”: I suggest exclude this phrase as that would be confusing to the reader.
- Line 305: “high scoring in community engagement to distil the elements these studies had in common; inferring the factors for success.” Again, is there a correlation between the studies with e.g., low mortality rate, high participation rate and high adherence to guidelines, and the community engagement? Are the studies that are successful in terms of quantifiable outcome measures also the studies that show high community engagement?
- Table 4: I wondered if the main conclusion from this table is that the two elements of success are community participation and incorporation of culture in research? Again, how is this success measured? The title of the table should describe what is seen is columns and rows. Also typo in “Success fact 2”
Conclusion
Line 329: Typo in “counties”, should be “countries”.
Author Response
Please see attachment and manuscript with tracked changes.

Reviewer 2 Report
Vita Christie and group have nicely reviewed safe therapies for breast cancer in indigenous women in high income countries. They systematically analyzed literature and have enlisted reason for setbacks faced by indigenous women with breast cancer. They pinpoint multiples barriers such culture, geographical, stigma, lack of appropriate knowledge. These thematic observations will definitely help to understand problem associated with breast cancer diagnosis and treatment. However, I find one shortcoming with this review. Authors have just diagnostic part and did not discuss in detail about treatment options and how they are affected by mentioned thematic barriers. Also discuss about strategies that have been implemented or will be implemented in future to overcome these thematic barriers.
Author Response

(The authors gave the same response as above.)

Round 2
Reviewer 2 Report
Authors have made appropriate changes to manuscript as per my suggestions. I find manuscript suitable for publication in current form.